# High-performance reconstruction of microscopic force fields from Brownian trajectories

Laura Pérez García[1], Jaime Donlucas Pérez[1], Giorgio Volpe [2], Alejandro V. Arzola[1] & Giovanni Volpe [3]

The accurate measurement of microscopic force fields is crucial in many branches of science and technology, from biophotonics and mechanobiology to microscopy and optomechanics. These forces are often probed by analysing their influence on the motion of Brownian particles. Here we introduce a powerful algorithm for microscopic force reconstruction via maximum-likelihood-estimator analysis (FORMA) to retrieve the force field acting on a Brownian particle from the analysis of its displacements. FORMA estimates accurately the conservative and non-conservative components of the force field with important advantages over established techniques, being parameter-free, requiring ten-fold less data and executing orders-of-magnitude faster. We demonstrate FORMA performance using optical tweezers, showing how, outperforming other available techniques, it can identify and characterise stable and unstable equilibrium points in generic force fields. Thanks to its high performance, FORMA can accelerate the development of microscopic and nanoscopic force transducers for physics, biology and engineering.

[1] Instituto de Física, Universidad Nacional Autónoma de México, Apdo. Postal 20-364, 01000Cd. México, Mexico. [2] Department of Chemistry, University College London, 20 Gordon Street, London WC1H 0AJ, UK. [3] Department of Physics, University of Gothenburg, 41296 Gothenburg, Sweden. Correspondence and requests for materials should be addressed to G.V. (email: g.volpe@ucl.ac.uk) or to A.V.A. (email: alejandro@fisica.unam.mx) or to G.V. (email: giovanni.volpe@physics.gu.se)

In many experiments in biology, physics, and materials science, a microscopic colloidal particle is used to probe local forces[1–4]; this is the case, for example, in the measurement of the forces produced by biomolecules, cells, and colloidal interactions. Often particles are held by optical, acoustic, or magnetic tweezers in a harmonic trapping potential with stiffness $k$ so that a homogeneous force acting on the particle results in a displacement $\Delta x$ from the equilibrium position and can therefore be measured as $k\Delta x$. To perform such measurement, it is necessary to determine the value of $k$, which is often done by measuring the Brownian fluctuations of the particle around its stable equilibrium position. This is achieved by measuring the particle position as a function of time, $x(t)$, and then using some calibration algorithms; the most commonly employed techniques are the potential[5], the power spectral density (PSD)[6], and the autocorrelation function (ACF)[7] analyses (see Methods for details). The first method samples the particle position distribution $\rho(x)$, calculates the potential using the Boltzmann factor, and then fits the value of $k$; this method requires a series of independent particle positions acquired over a time much longer than the system equilibration time to sample the probability distribution, and depends on the choice of some analysis parameters, such as the size of the bins. The latter two methods respectively calculate the PSD and ACF of the particle trajectory in the trap and fit them to their theoretical form to find the value of $k$; both methods require a time series of correlated particle positions at regular time intervals with a sufficiently short timestep $\Delta t$, and depend on the choice of some analysis parameters that determine how the fits are made.

Here we introduce a new powerful algorithm for microscopic force reconstruction via maximum-likelihood-estimator analysis (FORMA). FORMA exploits the fact that in the proximity of an equilibrium position the force field can be approximated by a linear form[4,8], and therefore, optimally estimated using a linear maximum-likelihood-estimator (MLE)[9,10]. FORMA has several advantages over the methods mentioned above. First, FORMA executes much faster because its algorithm is based on linear algebra for which highly optimised libraries are readily available. Second, it requires less data and therefore it converges faster and with smaller error bars. Third, it has less stringent requirements on the input data, as it does not require a series of particle positions sampled at regular time intervals or for a time long enough to reconstruct the equilibrium distribution. Fourth, it is simpler to execute and automatise because it does not have any analysis parameter to be chosen. Fifth, it probes simultaneously the conservative and non-conservative components of the force field. Finally, as it does not need to use the trajectory of a particle held in a potential, it can identify and characterise both stable and unstable equilibrium points in extended force fields, and therefore it is compatible with a broader range of possible scenarios where a freely diffusing particle is used as a tracer, e.g., in microscopy and rheology.

## Results

**FORMA in one-dimension.** To introduce the algorithm in the simplest one-dimensional (1D) situation, we start by considering a spherical microparticle of radius $R$ immersed in a liquid with viscosity $\eta$, at temperature $T$, and held in a harmonic confining potential of stiffness $k$. Experimentally, we have used a standard optical tweezers with a single-focused laser beam to create a harmonic trap; in this configuration, the motion along each dimension is independent and can be treated separately as effectively 1D[4]. The details of the experimental setup are described in the Methods and Supplementary Figure 1. Briefly, we have employed a focused laser beam (Gaussian profile, linear

polarisation, wavelength 532 nm, power at the sample 0.8 mW) and used it to trap a silica microsphere ($R = 0.48 \pm 0.02$ μm) in an aqueous solution. We have tracked the particle position using digital video microscopy[11] with a spatial resolution below 5 nm and with a frame frequency $f_s = 4504.5$ s$^{-1}$, corresponding to a sampling timestep $\Delta t = 0.222$ ms. Note that, as in the case of the PSD and ACF analyses, the sampling frequency needs to be at least about one order of magnitude greater than the characteristic trap frequency in order to obtain accurate results (see also Supplementary Figure 2). The corresponding overdamped Langevin equation is[12]:

$$\dot{x} = -\frac{k}{\gamma}x + \sqrt{2D}w, \qquad (1)$$

where $x(t)$ is the 1D particle position, $\gamma = 6\pi\eta R$, $D = k_B T/\gamma$, and $w(t)$ is a 1D white noise. We assume to have $N$ measurements of the particle displacement $\Delta x_n$ at position $x_n$ during a time $\Delta t_n$, where $n = 1,...,N$ (note that the time intervals do not need to be equal). Discretizing Eq. (1), we obtain that the average viscous friction force in the $n$-th time interval is

$$f_n = \gamma\frac{\Delta x_n}{\Delta t_n} = -kx_n + \sigma w_n, \qquad (2)$$

where $\sigma = \sqrt{\frac{2D\gamma^2}{\Delta t_n}}$ and $w_n$ is a Gaussian random number with zero mean and unit variance[12]. The central observation is that Eq. (2) is a linear regression model[9,10], whose parameters, $k$ and $\sigma$, can therefore be optimally estimated with a maximum-likelihood estimator from a series of observations of the dependent ($f_n$) and independent ($x_n$) variables, as schematically illustrated in Fig. 1. The detailed derivation is provided in the Methods for the general case. Wu et al.[13] have proposed a direct force measurement method based on taking the average of $f_n$ around each position of the force field; this approach requires many more samples than

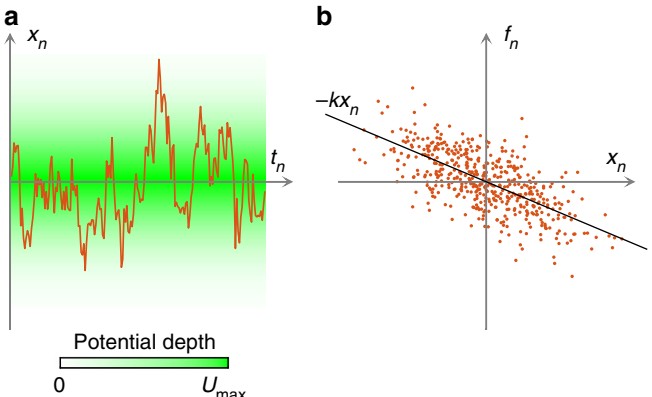

**Fig. 1** Force reconstruction via maximum-likelihood-estimator analysis. Schematic of the 1D version of FORMA for a particle held in an optical tweezers. **a** While a particle is held within a harmonic optical trap generated by an optical tweezers (the green background illustrates the depth of the potential), samples $x_n$ of its trajectory (solid orange line) are acquired at times $t_n$. **b** FORMA exploits the fact that the stiffness $k$ of the optical tweezers is related to the correlation between $x_n$ and the friction force $f_n$ acting on the particle (each dot represents a different ($x_n$, $f_n$) pair). FORMA uses an MLE to quickly, precisely, and accurately estimate this correlation and, thus, $k$. The spread of the dots around the linear regression line $-kx_n$ provides information about the diffusion coefficient $D$ of the particle

FORMA. The MLE estimation of the trap stiffness is then

$$k^* = \frac{\sum_n x_n f_n}{\sum_n x_n^2}. \tag{3}$$

Equation (3) is indeed a very simple expression that can be executed extremely fast using standard highly optimised linear algebra libraries, such as LAPACK[14], which is incorporated in most high-level programming languages, including MatLab and Python. Using the fact that $f_n + k^* x_n = \sigma\, w_n$ (Eq. 2), we can estimate the diffusion coefficient as

$$D^* = \frac{1}{N}\sum_n \frac{\Delta t_n}{2\gamma^2}[f_n + k^* x_n]^2 \tag{4}$$

and compare it to the expected value $D$, which provides an intrinsic quantitative consistency check for the quality of the estimation.

**Performance of FORMA.** In Fig. 2a, we show the estimation of the trap stiffness as a function of the number of samples (orange line). Already with as little as $10^3$ samples (corresponding to a total acquisition time of ~0.2 s), the stiffness has converged to its final value with small relative error (<20%), which improves as the number of samples increases reaching <2% relative error for $10^5$ samples (~20 s). The quality of the estimation can be evaluated by estimating the diffusion coefficient $D$ (Fig. 2b), which indeed converges to the expected value (dashed line) already for $10^3$ samples with a 3% relative error. Even for the highest number of samples, the algorithm execution time is in the order of a few milliseconds on a laptop computer (Fig. 2c). We have further verified these results on simulated data with physical parameters equal to the experimental ones (Fig. 2d–f). These simulations are in very good agreement with the results of the experiments and, given that the value of $k$ is fixed and known a priori (dashed line in Fig. 2d), they demonstrate the high accuracy of FORMA estimation: FORMA converges to the ground truth value of $k$ for about $10^4$ samples with 10% relative error, which, as in the experiments, reduces to 2% for $10^5$ samples.

In Fig. 2, we also compare the performance of FORMA with other established methods typically used in the calibration of optical tweezers, i.e. the potential, PSD, and ACF analyses[4–7] (the details and parameters used for these analyses are provided in the Methods). Overall, these results show that FORMA is more precise than other methods when estimating $k$ for a given number of samples, as the other methods typically need 10–100 times more data points to obtain comparable relative errors (Fig. 2a, b, d, and e). FORMA also executes faster by one to two orders of magnitude than the other methods (Fig. 2c, f). The relative errors of the FORMA estimation are also typically smaller, being 2% for $10^5$ samples, while the potential, PSD and ACF analyses achieve relative errors larger than 6, 7, and 20%, respectively. FORMA is also more accurate in estimating the value of $k$, as can been seen comparing experiments and simulations (Fig. 2a, d): whereas the potential and ACF analyses also converge to the correct $k$ value, the PSD analysis introduces a significant bias in the estimation (around 8%). Although for the specific case of the potential analysis (green lines), FORMA's performance can be considered a marginal improvement in terms of accuracy, it actually provides access to additional information that the potential analysis does not provide, namely the estimation of the diffusion coefficient $D$. Nonetheless, FORMA is significantly more accurate than the PSD analysis (blue lines), whose estimated $k$ and $D$ present a systematic error; with experience this error can be reduced by

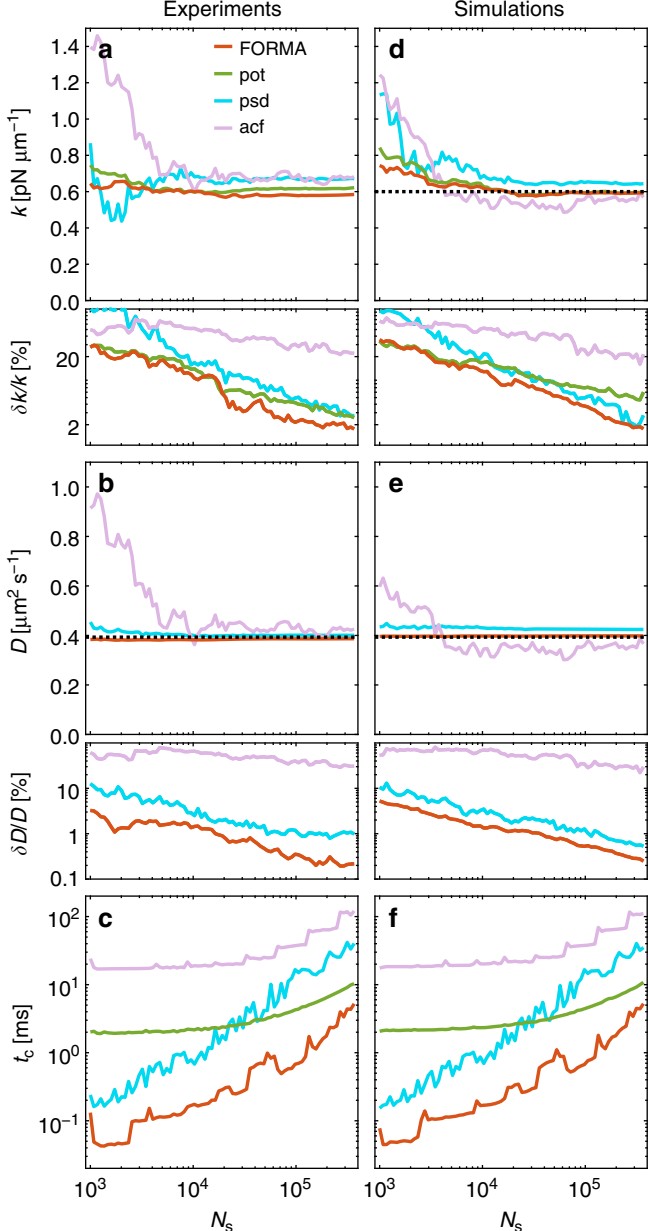

**Fig. 2** Better performance of FORMA compared to alternative techniques. Experimentally determined values of **a** the trap stiffness $k$ and its relative error $\delta k/k$, **b** the diffusion coefficient $D$ and its relative error $\delta D/D$, and **c** the computational execution time $t_c$ as a function of the sample number $N_s$ for FORMA (orange lines); and **d–f** corresponding results from numerical simulations. The comparisons with potential (green lines), PSD (blue lines), and ACF (pink lines) analyses show that FORMA converges faster (i.e. for smaller $N_s$), is more precise (i.e. smaller relative errors), is more accurate (i.e. it converges to the expected value represented by the black dashed line), and executes faster than the other methods. In all cases, we have acquired/simulated 24 trajectories of the motion of a spherical microparticle with radius $R = 0.48\ \mu m$ in an aqueous medium of viscosity $\eta = 0.0011\ Pa\ s^{-1}$ at a sampling frequency $f_s = 4504.5\ s^{-1}$. The relative errors are obtained as the standard deviations of the estimations over the 24 trajectories. The execution times are measured using a MatLab implementation of the algorithms on a laptop computer (processor Intel Core i7 at 2.2 GHz and 8 GB 1600 MHz DDR3)

tweaking the fitting PSD range, although this process can still be tricky without knowing a priori the value of $k$. Finally, FORMA is also significantly more precise and about two orders of magnitude faster than the ACF analysis (pink lines), which in fact is the least precise and the slowest method with 20% relative error and 100 ms execution time for $10^5$ samples, while FORMA has 2% relative error and 2 ms execution time for the same number of samples.

**FORMA in two-dimensions**. We now generalise FORMA to the two-dimensional (2D) case. Beyond a conservative component that is also present in 1D force fields, 2D force fields can also feature a non-conservative component[8]; as we will see, FORMA is able to estimate both simultaneously, differently from most other methods. The overdamped Langevin equation is now best written in vectorial form as

$$\dot{\mathbf{r}} = \frac{1}{\gamma}\mathbf{F}(\mathbf{r}) + \sqrt{2D}\,\mathbf{w}, \qquad (5)$$

where $\mathbf{F}(\mathbf{r})$ is a force field and $\mathbf{w}$ is a vector of independent white noises. $\mathbf{F}(\mathbf{r})$ can be expanded in Taylor series around $\mathbf{0}$ as $\mathbf{F}(\mathbf{r}) = \mathbf{F}_0 + \mathbf{J}_0\mathbf{r} + o(\mathbf{r})$, where $\mathbf{F}_0 = \mathbf{F}(\mathbf{0})$ is the force and $\mathbf{J}_0 = \mathbf{J}(\mathbf{0})$ the Jacobian at $\mathbf{0}$. If we assume that $\mathbf{0}$ is an equilibrium point, then $\mathbf{F}_0 = \mathbf{0}$; with this assumption, the results in the following become much simpler without loss of generality, as this is equivalent to translating the experimental reference frame so that it is centred at the equilibrium position (we discuss below how to proceed if the equilibrium position is not known). Analogously to the 1D case explained above, using Eq. (5) the average friction force in the $n$-th time interval is

$$\mathbf{f}_n = \gamma\frac{\Delta\mathbf{r}_n}{\Delta t_n} = \mathbf{J}_0\mathbf{r}_n + \sigma\mathbf{w}_n, \qquad (6)$$

where $\mathbf{w}_n$ is an array of independent random numbers with zero mean and unit variance. Eq. (6) is again a linear regression model and therefore the MLE estimator of $\mathbf{J}_0$ is given by

$$\mathbf{J}_0^* = \left[\mathbf{r}^\mathrm{T}\mathbf{r}\right]^{-1}\mathbf{r}^\mathrm{T}\mathbf{f}, \qquad (7)$$

where $\mathbf{r} = (\mathbf{r}_n)$ and $\mathbf{f} = (\mathbf{f}_n)$ are matrices with $N \times 2$ elements. Eq. (6) can be computed extremely efficiently as it only requires matrix multiplications and the trivial inversion of a $2 \times 2$ matrix. As in the 1D case, we can calculate the residual error (see Methods) and use it to determine the quality of the reconstruction of the force field by estimating the value of the diffusion coefficient along each of the two axes.

A schematic of the workflow of the 2D version of FORMA is presented in Fig. 3a–c. The estimated force field around the equilibrium point is $\mathbf{F}^*(\mathbf{r}) = \mathbf{J}_0^*\,\mathbf{r}$, where we use the estimated Jacobian[8] (Eq. 7). This is a linear form that results from the superposition of a conservative harmonic potential (which is characterised by its stiffnesses $k_1^*$ and $k_2^*$ along the principle axes, and the orientation $\theta^*$ of the principle axes with respect to the Cartesian axes) and a non-conservative rotational force field (which is characterised by its angular frequency $\Omega^*$). Some examples of this decomposition are shown in Supplementary Figure 3. It is possible to obtain these two components directly from the Jacobian, by separating it into its conservative and non-conservative parts as $\mathbf{J}_0^* = \mathbf{J}_c^* + \mathbf{J}_r^*$, making use of the fact that they are respectively symmetric and antisymmetric. The conservative part is

$$\mathbf{J}_c^* = \frac{1}{2}(\mathbf{J}_0^* + \mathbf{J}_0^{*\mathrm{T}}) = \mathbf{R}(\theta^*)\begin{bmatrix} -k_1^* & 0 \\ 0 & -k_2^* \end{bmatrix}\mathbf{R}^{-1}(\theta^*), \qquad (8)$$

where $\mathbf{R}(\theta)$ is a rotation matrix that diagonalises $\mathbf{J}_c$ and whose

principal axes correspond to the eigenvectors corresponding to the principle axes of the harmonic potential and the stiffnesses along these axes correspond to the eigenvalues with a minus sign. The non-conservative (rotational) part is

$$\mathbf{J}_r^* = \frac{1}{2}(\mathbf{J}_0^* - \mathbf{J}_0^{*\mathrm{T}}) = \begin{bmatrix} 0 & -\gamma\Omega \\ \gamma\Omega & 0 \end{bmatrix} \qquad (9)$$

and, as it is invariant under a rotation of the reference system, the angular frequency can be simply estimated as

$$\Omega^* = \frac{1}{2\gamma}\left[J_{0,21}^* - J_{0,12}^*\right]. \qquad (10)$$

To demonstrate this 2D version of FORMA at work, we have used it to estimate the transfer of orbital and spin angular momentum to an optically trapped particle. In fact, orbital and spin angular momentum can make a transparent particle orbit, even though the precise angular-momentum-transfer mechanisms can be very complex when the beam is tightly focused, depending on the size, shape and material of the particle, as well as on the size and shape of the beam[13,15–19]. We employ the same setup and microparticle as for the results presented in Fig. 2 (see Methods and Supplementary Figure 1), using a spatial light modulator to generate Laguerre-Gaussian (LG) beams carrying orbital angular momentum (OAM) to trap the particle and a quarter-wave plate to switch their polarisation state from linear to positive or negative circular polarisation. The results of the force field reconstruction are the stiffnesses $k_x$ and $k_y$ (Fig. 3d–f) and the angular frequencies $\Omega$ (Fig. 3g–i) of a Brownian particle optically trapped in various LG beams. In Fig. 3d, g, we employ a linearly polarised LG beam with topological charge $l = -2, -1, 0, 1, 2$ (the case $l = 0$ corresponds to the standard Gaussian beam already employed in Fig. 2). As already observed in previous experiments, we also measure that $\Omega^*$ is proportional to $l$ ($\Omega^* \approx 1.8\,\mathrm{s}^{-1}l$), which follows the quantisation of the OAM[20]. We have verified that these values are in good quantitative agreement with those obtained using the more standard cross-correlation function (CCF) analysis, which is an extension of the ACF analysis that permits one to detect the presence of non-conservative force fields[8]. These non-conservative rotational force fields are very small, produce an almost imperceptible bending of the force lines when comparing $\mathrm{LG}_0$ with $\mathrm{LG}_2$ (insets of Fig. 3g), and, therefore, cannot be detected by directly counting rotations of the particle around the beam axis. To further test our results, we have then changed the polarisation state of the beam switching it to positive circular polarisation (Fig. 3e, h), which introduces an additional spin angular momentum (SAM), so that $\Omega^* \approx 1.8\,\mathrm{s}^{-1}(l+1)$, recovering the SAM quantisation. This relation suggests that the OAM and SAM contribute equally to the rotation of the particle. We finally changed the polarisation state of the beam to negative circular polarisation (Fig. 3f, i) obtaining $\Omega^* \approx 1.8\,\mathrm{s}^{-1}(l-1)$.

**FORMA in three-dimensions**. Following the same procedure, FORMA can be extended also to measure three-dimensional (3D) force fields. The resulting equations are provided in the Methods, and we have tested them on simulated data (see Supplementary Table 1).

**Measurement of stable and unstable equilibrium points**. Until now, we have always centred the reference frame at the equilibrium position. However, the positions of the equilibria might be unknown a priori, for example when exploring extended potential landscapes. FORMA can be further refined to address this

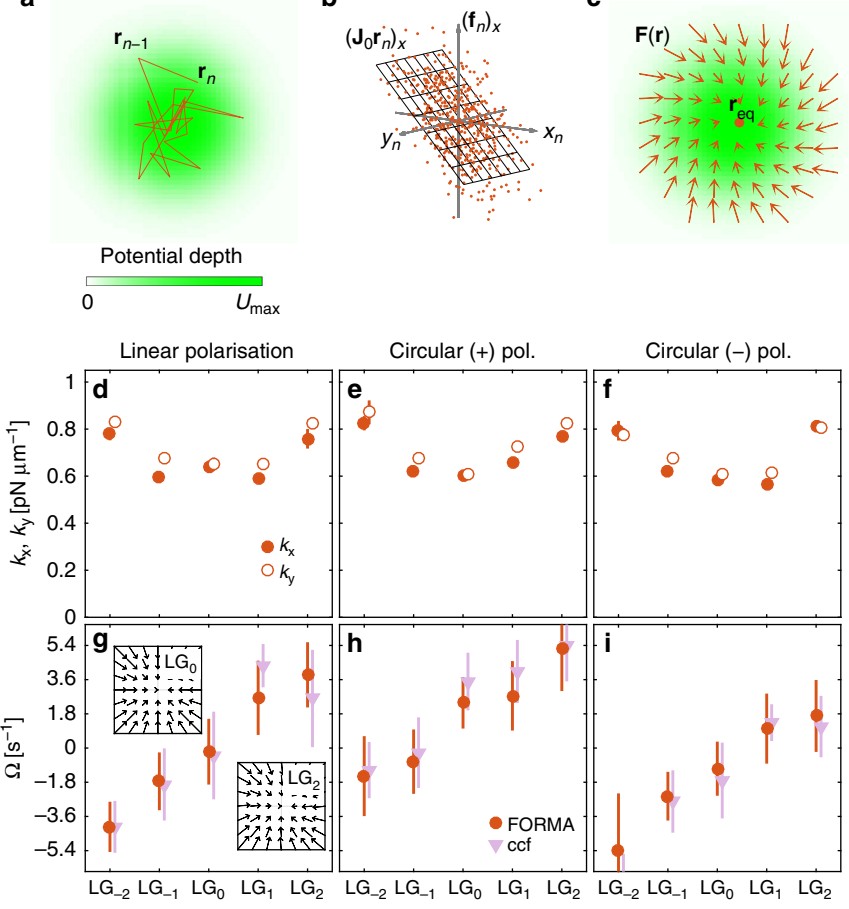

**Fig. 3** Measurement of the non-conservative force-field component. **a–c** Schematic of the 2D version of FORMA for a particle held in an optical tweezers: **a** Samples $\mathbf{r}_n$ of a particle trajectory (solid orange line) held in an optical tweezers (the green background illustrates the depth of the potential) are acquired at times $t_n$. **b** FORMA estimates the Jacobian $\mathbf{J}_0$ of the force field from the relation between $\mathbf{r}_n$ and $\mathbf{f}_n$ using a 2D MLE. In the schematic, we represent only the estimation of the first row of $\mathbf{J}_0$, which is related to the $x$-component of $\mathbf{f}_n$; the complete graph cannot be represented because it is 4D. **c** Using this information, FORMA reconstructs the force field around the equilibrium point $\mathbf{r}_{eq}$ (see also Supplementary Figure 3). **d–f** Stiffnesses $k_x$ and $k_y$, and **g–i** angular frequency $\Omega$ of a Brownian particle optically trapped by **d**, **g** a linearly polarised, **e**, **h** circularly (+) polarised, and **f**, **i** circularly (−) polarised Laguerre-Gaussian (LG) beam with $l = -2, -1, 0, 1, 2$. **g–i** The results of FORMA (orange circles) agree well with the results of the CCF analysis (pink triangles). The insets in **g** show the force fields for the case of a Gaussian beam (LG$_0$), which is purely conservative ($l = 0$), and of a beam with a charge $l = 2$ of orbital angular moment (LG$_2$), which features a non-conservative component that only induces a mild bending of the arrows. In all cases, we have acquired trajectories of the motion of a spherical particle with radius $R = 0.48\,\mu m$ in an aqueous medium of viscosity $\eta = 0.0011\,Pa\,s^{-1}$ at a sampling frequency $f_s = 4504.5\,s^{-1}$, and used 25 windows of $10^5$ samples for the analysis; the error bars in **d–i** are the standard deviations over these 25 measurements

problem and determine the value of $\mathbf{F}_0$ in Eq. (5), which permits one to identify equilibrium points when $\mathbf{F}_0 = \mathbf{0}$. We obtain the following estimator (see Methods for detailed derivation):

$$\left[\mathbf{J}_0^* \mathbf{F}_0^*\right] = \left(\widetilde{\mathbf{r}}^{\mathrm{T}}\widetilde{\mathbf{r}}\right)^{-1}\widetilde{\mathbf{r}}^{\mathrm{T}}\mathbf{f}, \tag{11}$$

where $\widetilde{\mathbf{r}} = [\mathbf{r}\,\mathbf{1}]$ with $\mathbf{1}$ a column vector constituted of $N$ ones. Having the trajectory of a particle moving in an extended potential landscape, it is possible to use Eq. (11) to simultaneously identify the equilibrium points, classify their stability, and characterise their local force field: For each position in the potential landscape, the parts of the trajectory that fall within a radius $a$ smaller than the characteristic length over which the force field varies from this position are selected and analysed with FORMA; if $\mathbf{F}_0^* \approx \mathbf{0}$, then this position is an equilibrium point and $\mathbf{J}_0^*$ permits us to determine the local force field.

We have applied this procedure to identify the equilibrium points in a multistable potential, which we realised focusing two slightly displaced laser beams obtained using a spatial light modulator (see Methods and Supplementary Figure 1). Similar

configurations have been extensively studied as a model system for thermally activated transitions in bistable potentials[21,22]; however, the presence of additional minima other than the two typically expected has been recognized only recently due to their weak elusive nature[23]. The results of the reconstruction are shown in Fig. 4. In Fig. 4a, we use the potential analysis to determine the potential (green solid line with error bars denoted by the shaded area) by acquiring a sufficiently long trajectory so that the particle has equilibrated and fully explored the region of interest; this potential appears to be bistable with two potential minima (stable equilibrium points) and a potential barrier between them corresponding to an unstable equilibrium point. When we use FORMA, we identify five equilibria, and classify them as stable ($x_1^*, x_3^*, x_5^*$; full circles) and unstable ($x_2^*, x_4^*$; empty circles); importantly, we are able to clearly resolve the presence of additional equilibrium points within the potential barrier ($x_2^*, x_3^*, x_4^*$). For each point, FORMA provides the respective stiffness (the corresponding harmonic potentials are shown by the orange lines), which is negative for unstable equilibrium points. This highlights one of the additional key advantages of FORMA: It can

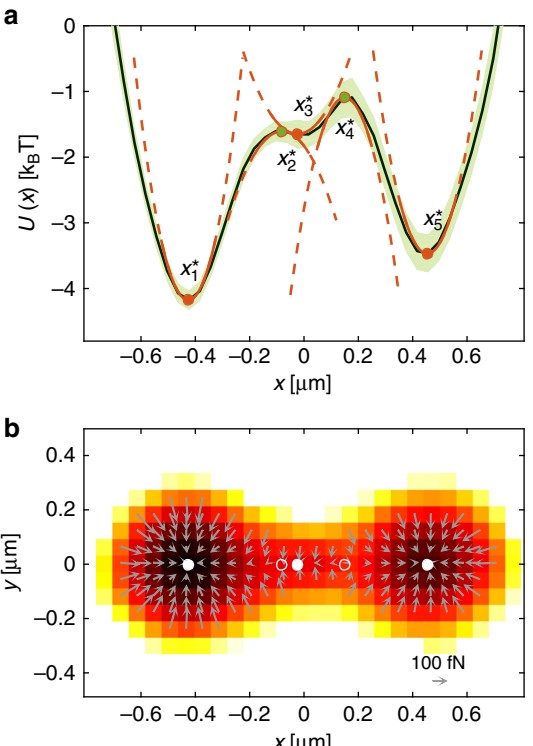

**Fig. 4** Reconstruction of stable and unstable equilibrium points. **a** Multiwell optical potential generated by two focused Gaussian beams slightly displaced along the x-direction. FORMA identifies three stable ($x_1^*$, $x_3^*$, $x_5^*$; full circles) and two unstable ($x_2^*$, $x_4^*$; empty circles) equilibrium points, and measures their stiffness (orange solid and dashed lines). The corresponding x-potential obtained from the potential method is shown by the green solid line (the green shaded area represents one standard deviation obtained repeating the experiment 20 times). **b** 2D plot of the force field measured with FORMA (arrows) and of the potential measured with the potential analysis (background colour). The stable and unstable equilibrium points are indicated by the full and empty circles, respectively. We have acquired trajectories of the motion of a spherical particle with radius $R = 0.48\,\mu m$ in an aqueous medium of viscosity $\eta = 0.0011\,Pa\,s^{-1}$ at a sampling frequency $f_s = 4504.5\,s^{-1}$, and used 20 windows of $4.5 \times 10^5$ samples for the analysis

also measure the properties of the force field around unstable equilibrium points, thanks to the fact that it does not require an uninterrupted series of data points or a complete sampling of the equilibrium distribution, which are difficult or impossible near an unstable equilibrium. Figure 4b shows a 2D view of the potential, where we used FORMA to estimate the force field on a 2D grid and to identify the stable and unstable equilibria. The background colour represents the potential reconstructed using the equilibrium distribution, which shows a good agreement with the results of FORMA, but does not allow to clearly identify the presence of the additional equilibria in the potential barrier.

**Measurement of extended force fields.** Finally, we can also use FORMA to study larger extended force fields, such as the random optical force fields generated by speckle patterns[24–28]. Speckles are complex interference patterns with well-defined statistical properties generated by the scattering of coherent light by disordered structures[29]; the equilibrium positions are not known a priori due to their random appearance, the configuration space is virtually infinite, and there can be a non-conservative component. Thus, the challenge is at least twofold. First, the configuration space is virtually infinite and, therefore, cannot be sampled by a single trajectory in any reasonable amount of time. Second, the

force field can present a non-conservative component; in fact, whereas several works have demonstrated micromanipulation with speckle patterns[24–28], none of them has so far achieved the experimental characterisation of this non-conservative nature of the optical forces. To study this situation, we have employed a speckle light field generated using a second optical setup (see Methods and Supplementary Figure 4). A portion of the resulting speckle field is shown by the green background in Fig. 5a. To sample the force field, we have acquired the trajectories of a particle in various regions of the speckle field: In each of these trajectories the particle typically explores the regions surrounding several contiguous stable equilibrium points by being metastably trapped in each of them while still being able to cross over the potential barriers separating them[30]. These trajectories cannot be used in the potential analysis because they do not provide a fair sampling of the position space. Nevertheless, they can be used by FORMA to identify the equilibrium points, which are shown in Fig. 5a by the full circles (stable points) and empty circles (unstable and saddle points), and to determine the force field around them (see Supplementary Table 2 for the measured values): For example, in Fig. 5b we show a stable point, in Fig. 5c an unstable point with a significant rotational component, and in Fig. 5d a series of two stable points with a saddle between them in a configuration reminiscent of that explored in Fig. 4.

**Discussion**

We have introduced FORMA: a new, powerful algorithm to measure microscopic force fields using the Brownian motion of a microscopic particle based on a linear MLE. We first introduced the 1D version of FORMA; we quantitatively compared it to other standard methods, showing that it needs less samples, it has smaller relative errors, it is more accurate, and it is orders-of-magnitude faster. We then introduced the 2D version of FORMA; we showed that it can also measure the presence of a non-conservative force field, going beyond what can be done by the other methods. Finally, we applied it to more general force-field landscapes, including situations where the shape of the potential is not harmonic and the forces are too shallow to achieve long-term trapping: we used FORMA to identify the equilibrium points; to classify them as stable, unstable and saddle points; and to characterise their local force fields. Overall, we have shown that, requiring less data and having a faster run time, FORMA can be applied to situations that require a fast response such as in real-time applications (e.g., in haptic optical tweezers[31]) and in the presence of time-varying conditions (e.g., in the study of biological systems[32–34], of active baths[35,36] and of systems out of equilibrium[37–39]). Even though we have presented results only for particles of a single size, FORMA can be used as described in this work for spherical particles of any size, from Rayleigh particles much smaller than the optical wavelength to large Mie particles. FORMA can be straighforwardly extended also to measure flow fields and 3D force fields. FORMA can also be extended to deal with non-translational degrees of freedom, which might be important when dealing with non-spherical particles. For force fields that cannot be approximated by a linear form around an equilibrium position, because $\mathbf{J}_0 = \mathbf{0}$, FORMA needs to be extended using a higher-order MLE estimator. Thanks to the fact that this algorithm is significantly faster, simpler and more robust than commonly employed alternatives, it has the potential to accelerate the development of force transducers capable of measuring and applying forces on microscopic and nanoscopic scales, which are needed in many areas of physics, biology, and engineering.

**Methods**

**Potential analysis.** The potential method[4,5] relies on the fact that a harmonic potential has the form $U(x) = \frac{1}{2} kx^2$ and the associated position probability

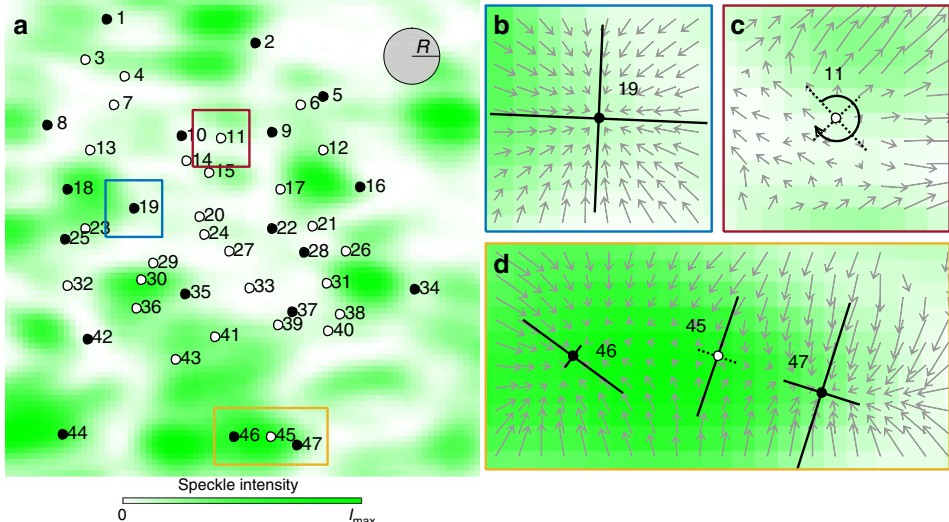

**Fig. 5** Reconstruction of the equilibrium points in a speckle pattern. **a** The intensity of the speckle (green background, laser wavelength $\lambda = 532$ nm) is approximately proportional to the potential depth of the optical potential felt by a particle whose size (particle diameter $1.00 \pm 0.04$ μm) is similar to the speckle characteristic size (2.8 μm)[27, 30]. FORMA identifies several stable (full circles) and unstable (empty circles) equilibrium points, and measures the orientation of the principal axes ($\theta$), the stiffness along them ($k_1$ and $k_2$), and angular frequency ($\Omega$) (see Supplementary Table 2 for the measured values). We have acquired trajectories of the motion of a spherical particle with radius $R = 0.50 \pm 0.02$ μm in an aqueous medium of viscosity $\eta = 0.0013$ Pa s$^{-1}$ at a sampling frequency $f_s = 600$ s$^{-1}$; as we cannot expect the particle to spontaneously diffuse over the whole speckle field during the time of the experiment, we have placed this particle in 25 positions within the speckle field and let it diffuse each time acquiring $2 \times 10^6$ samples for the analysis. **b–d** Examples of reconstructed force fields around **b** a stable point, **c** an unstable point with a significant rotational component (indicated by the arrow), and **d** two stable points with a saddle in between; the grey arrows plot the 2D force field measured with FORMA and are scaled by a different factor in each plot

distribution of the particle is $\rho(x) = \rho_0 \exp\left(-\frac{U(x)}{k_B T}\right)$. Therefore, by sampling $\rho(x)$ it is possible to reconstruct $U(x) = -k_B T \ln(\rho(x))$ and $F(x) = -\frac{dU(x)}{dx}$. The value of $k$ is finally obtained by fitting $F(x)$ to a linear function. The potential method requires a series of particle positions acquired over a time long enough for the system to equilibrate as well as the choice of the number of bins and of the fitting algorithm to be employed in the analysis: here, we used 100 bins equally spaced between the minimum and maximum value of the particle position, and a linear fitting.

**Power spectral density analysis**. The PSD method[4,6] uses the particle trajectory in the harmonic trap, $x(t)$, to calculate the PSD, $P(f) = \frac{D}{2\pi^2}(f_c^2 + f^2)^{-1}$, where $f_c = (2\pi\gamma)^{-1}k$ is the harmonic trap cutoff frequency, $\gamma$ is the particle friction coefficient, and $D$ is its diffusion coefficient. It then fits this function to find the value of $k$, and therefore the harmonic force field surrounding the particle. The PSD method requires a time series of correlated particle positions at regular time intervals with a sufficiently short timestep $\Delta t$. It also requires to choose how the PSD is calculated (e.g., use of windowing and binning[6]) and the frequency range over which the fitting is made: here, we performed the PSD fitting over the frequency range between five times the minimum measured frequency and half the Nyquist frequency without using windowing and binning.

**Auto-correlation function analysis**. The ACF method[4,7] calculates the ACF of the particle position, $C(\tau) = \frac{k_B T}{k} \exp\left(-\frac{k|\tau|}{\gamma}\right)$, where $k_B$ is the Boltzmann constant and $T$ is the absolute temperature. It then fits this function to find the value of $k$, and therefore the harmonic force field surrounding the particle. Like the PSD method, also the ACF method requires a time series of correlated particle positions at regular time intervals. It also requires to choose over which range to perform the fitting: here, we have performed the fitting over the values of the ACF >1% its maximum.

**Experimental setups**. For the single-beam (Figs. 2 and 3) and two-beam (Fig. 4) experiments, we used the standard optical tweezers shown in Supplementary Figure 1[4,28,40]. An expanded 532-nm-wavelength laser beam (power at the sample 0.8 mW) is reflected by a spatial light modulator (SLM) in a 4f-configuration with a diaphragm in the Fourier space acting as spatial filter[4,28]. By altering the phase profile of the beam we generate different beams, including the Gaussian beam used in Fig. 2, the Laguerre-Gaussian beams with $l = -2, -1, 0, 1, 2$ used in Fig. 3, and the double beam used in Fig. 4. We control the polarization of the beam using a

quarter-wave plate, which permits us to switch the polarisation state of the beam between linearly polarised (Fig. 3d, g), circularly (+) polarised (Fig. 3e, h), and circularly (−) polarised (Fig. 3f, i). These experiments are performed with spherical silica microparticles with radius $R = 0.48 \pm 0.02$ μm in an aqueous medium of viscosity $\eta = 0.0011$ Pa s$^{-1}$ (corrected using Faxén formula for the proximity of the cover slip[41]) whose position is acquired with digital video microscopy[11] at a sampling frequency $f_s = 4504.5$ s$^{-1}$ and a resolution below 5 nm.

For the speckle experiments (Fig. 5), we used the SLM in the image plane, as shown in Supplementary Figure 4: The 532-nm laser beam is reflected by the SLM, which projects a random phase (with uniform distribution of values in (0, $2\pi$)) in every domain of $6 \times 6$ pixels, and is directed to the sample using two telescopes. We control the effective numerical aperture of the system, and therefore the grain size of the speckle, by using a diaphragm with a diameter $D_i = 1.54 \pm 0.05$ mm in the Fourier plane of the first telescope. The mean intensity of the speckle is 0.015 mW μm$^{-2}$. These experiments are performed with spherical polystyrene microparticles with radius $R = 0.50 \pm 0.02$ μm in an aqueous medium of viscosity $\eta = 0.0013$ Pa s$^{-1}$ measured by using the mean diffusion of the particle in the speckle pattern obtained from FORMA and using Einstein-Stokes relation. Because of the radiation pressure of the optical forces generated by the speckle light field, the probe bead is pushed towards the upper cover slip (without touching it because of screened electrostatic repulsion), where it remains confined in a quasi-2D configuration and diffuses exploring a wide area. In order to have control of the region of interest the initial positions of the particles were prepared using an optical trap generated with the same laser beam and SLM. The particle's position is recorded for $2 \times 10^6$ frames each at a sampling frequency $f_s = 600$ s$^{-1}$.

**Detailed derivation of FORMA in two-dimensions**. Here we derive FORMA in its most general form presented in the article (Eq. 11). The average friction force in the $n$-th time interval is

$$\mathbf{f}_n = \gamma \frac{\Delta \mathbf{r}_n}{\Delta t_n} = \mathbf{F}_0 + \mathbf{J}_0 \mathbf{r}_n + \sqrt{\frac{2k_B T \gamma}{\Delta t_n}} \mathbf{w}_n, \tag{12}$$

which can be rewritten explicitly as

$$\begin{bmatrix} f_{x,n} \\ f_{y,n} \end{bmatrix} \approx \begin{bmatrix} J_{0,11} & J_{0,12} & F_{0,x} \\ J_{0,21} & J_{0,22} & F_{0,y} \end{bmatrix} \begin{bmatrix} x_n \\ y_n \\ 1 \end{bmatrix} + \sqrt{\frac{2k_B T \gamma}{\Delta t}} \begin{bmatrix} w_{x,n} \\ w_{y,n} \end{bmatrix}, \tag{13}$$

Assuming to have $N$ measurements, we introduce the vectors

$$\mathbf{f} = \begin{bmatrix} \gamma\frac{\Delta x_1}{\Delta t_1} & \gamma\frac{\Delta y_1}{\Delta t_1} \\ \dots & \dots \\ \gamma\frac{\Delta x_n}{\Delta t_n} & \gamma\frac{\Delta y_n}{\Delta t_n} \\ \dots & \dots \\ \gamma\frac{\Delta x_N}{\Delta t_N} & \gamma\frac{\Delta y_N}{\Delta t_N} \end{bmatrix} \qquad (14)$$

and

$$\widetilde{\mathbf{r}} = \begin{bmatrix} x_1 & y_1 & 1 \\ \dots & \dots & \dots \\ x_n & y_n & 1 \\ \dots & \dots & \dots \\ x_N & y_N & 1 \end{bmatrix}, \qquad (15)$$

the MLE is given by Eq. (11), i.e.,

$$\begin{bmatrix} \mathbf{J}_0^* & \mathbf{F}_0^* \end{bmatrix} = \begin{bmatrix} J_{0,11}^* & J_{0,12}^* & F_{0,x}^* \\ J_{0,21}^* & J_{0,22}^* & F_{0,y}^* \end{bmatrix} = (\widetilde{\mathbf{r}}^{\mathrm{T}}\widetilde{\mathbf{r}})^{-1}\widetilde{\mathbf{r}}^{\mathrm{T}}\mathbf{f}$$

and the estimated particle diffusivity along each axis can be calculated from the residual error of the MLE

$$\begin{aligned} D_x^* &= \frac{1}{N}\sum_{n=1}^{N}\frac{\Delta t_n}{2\gamma^2}\left(f_{x,n} - J_{0,11}^* x_n - J_{0,12}^* y_n - F_{0,x}^*\right)^2 \\ D_y^* &= \frac{1}{N}\sum_{n=1}^{N}\frac{\Delta t_n}{2\gamma^2}\left(f_{y,n} - J_{0,21}^* x_n - J_{0,22}^* y_n - F_{0,y}^*\right)^2 \end{aligned} \qquad (16)$$

**Codes**. We provide the MatLab implementations of the key functionalities of FORMA[42].

For the 1D version of FORMA, we provide the following codes: function forma1d.m, script test_forma1d.m to execute it, and set of test data forma1d.mat. This code estimates the values of $k^*$ and $D^*$ assuming that the equilibrium position is at $x = 0$ and implementing Eqs. (3) and (4).

For the 2D version of FORMA, we provide the following codes: function forma2d.m, script test_forma2d.m to execute it, and set of test data forma2d.mat. This code estimates the value of $k_1^*$, $k_2^*$, $\theta^*$, and $\Omega^*$ assuming that the equilibrium position is at $\mathbf{r}_0 = \mathbf{0}$ and implementing Eqs. (7), (8), and (10).

**FORMA in three-dimensions**. Following the same steps as in the previous sections, the MLE of a particle in 3D near an equilibrium position can be explicitly written as

$$\mathbf{J}_0^* = \begin{bmatrix} J_{0,11}^* & J_{0,12}^* & J_{0,13}^* \\ J_{0,21}^* & J_{0,22}^* & J_{0,23}^* \\ J_{0,31}^* & J_{0,32}^* & J_{0,33}^* \end{bmatrix} = (\widetilde{\mathbf{r}}^{\mathrm{T}}\widetilde{\mathbf{r}})^{-1}\widetilde{\mathbf{r}}^{\mathrm{T}}\mathbf{f}, \qquad (17)$$

where $\mathbf{f}$ and $\widetilde{\mathbf{r}}$ are the corresponding 3D extensions of Eqs. (14) and (15). The estimated particle diffusivity along each axis can be calculated from the residual error of the MLE

$$\begin{aligned} D_x^* &= \frac{1}{N}\sum_{n=1}^{N}\frac{\Delta t_n}{2\gamma^2}\left(f_{x,n} - J_{0,11}^* x_n - J_{0,12}^* y_n - J_{0,13}^* z_n\right)^2, \\ D_y^* &= \frac{1}{N}\sum_{n=1}^{N}\frac{\Delta t_n}{2\gamma^2}\left(f_{y,n} - J_{0,21}^* x_n - J_{0,22}^* y_n - J_{0,23}^* z_n\right)^2, \\ D_z^* &= \frac{1}{N}\sum_{n=1}^{N}\frac{\Delta t_n}{2\gamma^2}\left(f_{z,n} - J_{0,31}^* x_n - J_{0,32}^* y_n - J_{0,33}^* z_n\right)^2. \end{aligned} \qquad (18)$$

Using Brownian simulations, we were able to prove the performance of this method in 3D. The data in the Supplementary Table 1 show the results of the analysis of an hypothetic optical trap with defined properties along the three axis.

## Data availability

Data and codes supporting the findings of this study are available in figshare with the digital object identifier https://doi.org/10.6084/m9.figshare.7181888[42]. Further data and resources in support of the findings of this study are available from the corresponding authors upon reasonable request.

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

## Acknowledgements

We thank Karen Volke-Sepulveda for useful discussions and Antonio A. R. Neves for critical reading of the manuscript. L.P.G., J.D., and A.V.A. acknowledge funding from DGAPA-UNAM (grants PAPIIT IA104917 and IN114517). A.V.A. and G.V. (Giovanni Volpe) acknowledge support from Cátedra Elena Aizen de Moshinsky. G.V. (Giorgio Volpe) acknowledges support from the Wellcome Trust [204240/Z/16/Z]. A.V.A. acknowledges the Swedish Council for Higher Education through the Linnaeus-Palme International Exchange Program (contract 3.3.1.34.10235-2018). G.V. (Giovanni Volpe) was partially supported by the ERC Starting Grant ComplexSwimmers (Grant No. 677511).

## Author contributions

G.V. (Giovanni Volpe) had the original idea for this method while visiting the National Autonomous University of Mexico (UNAM). L.P.G. and A.V.A. performed most of the experiments and analysed the data; J.D.L. performed the experiments reported in Fig. 4. G.V. (Giorgio Volpe) and G.V. (Giovanni Volpe) performed the simulations. L.P.G., G.V. (Giorgio Volpe), A.V.A. and G.V. (Giovanni Volpe) discussed the data and wrote the draft of the article. All authors revised the final version of the article.

## Additional information

**Competing interests:** The authors declare no competing interests.

