## [Peer Review file · Nature Communications]

Reviewers' comments:

Reviewer #1 (Remarks to the Author):

Optical trapping remains an important field with continued success in a variety of areas in mesoscopic physics. One of the key requirements of the area is a determination of forces which is associated with a deeper understanding of the Brownian dynamics and optical fields interacting with the trapped object.

This manuscript develops a new algorithm for microscopic Force Reconstruction using a Maximum-likelihood-estimator (MLE) Analysis (FORMA) to retrieve the force field acting on a Brownian particle from the analysis of its displacements. This approach of FORMA allows the authors to make precise simultaneous estimations of both the conservative and non-conservative components of the force field. Importantly, Even for the highest number of samples, the algorithm run time is in the order of a few ms on a laptop making this approach very attractive. The performance of FORMA is benchmarked with other established methods typically used in the calibration of optical tweezers the potential, PSD, and ACF analyses and studies performed with light fields with spin and orbital angular momentum and for particles in speckle for particles $\sim 1\mu\text{m}$ in diameter.

I would remark that non-conservative fields have come to the fore recently due to investigations in both standard traps and also for particles trapped in – for example – optical vortices where scattering of light can transfer orbital angular momentum.

Overall, this is an interesting, topical paper that is likely to find interest in the trapping community and potentially beyond. In parts it is well written and clear. Some questions for the authors

1) FORMA relies on the fact that the proximity of an equilibrium position the force field can be approximated by a linear form leading to the use of use of a linear MLE. How accurate would this be if we move away dramatically from harmonic/Gaussian forms of traps or even have highly aberrated/distorted traps – e.g. a trap inside a cell? When does the assumptions made here breakdown?

2) Can FORMA truly extract 3D information of a single particle or even be used for multiple trapped particles at different x,y,z positions? The authors seem to fleetingly state: FORMA can be straightforwardly extended also to measure....3D force fields."

This is insufficient and needs data to support the 3D nature of the approach and some detailed discussion. It should be included to aid fair comparison to the other methods used in optical tweezers. At present this appears a shortcoming of the approach. See next point too which shows a method for 3D force field determination.

3) The following paper shows relevant work and is not cited nor discussed in context. The paper extracts local forces from trajectories of an optically trapped particle, and reveals the three-dimensional force field experienced by a Rayleigh particle with 10 nm spatial resolution and femtonewton precision in force.

Direct Measurement of the Nonconservative Force Field Generated by Optical Tweezers
Pinyu Wu, Rongxin Huang, Christian Tischer, Alexandr Jonas, and Ernst-Ludwig Florin
Phys. Rev. Lett. 103, 108101 (2009)

5) There are several English grammar that might be clarified and errors in references

"the precise angular-momentum-transfer mechanisms can be very complex when the beam is focused.." – should be "...when the beam is tightly focused"

e.g. ref 13, 19 seem incomplete;

Figure 3: if the y-axis is angular velocity should it not be rad/s and not just per second?

6) is there any drawback to using non-spherical particles or Rayleigh particles with FORMA? All data presently shown is for larger spherical objects.

7) The authors state: " FORMA can be applied in situations that require a fast response such as in real-time applications and in the presence of time-varying conditions." Is there any such specific situation they can show where FORMA works but the other methods do not?

8) I don't get any sense of the limits of FORMA at present. For example the determination of stiffness using the power spectrum typically requires a harmonic potential, a sampling bandwidth well in excess of the trap frequency, a detailed knowledge of the drag coefficient, and some form of accurate position calibration. The authors have focused on speed in the paper rather than other aspects of FORMA. They should expand this aspect.

Overall, I think there may be potential for this work for publication in Nature Communications but the authors first need to provide a detailed response to the above issues

Reviewer #2 (Remarks to the Author):

This is a nice piece of work and I'm confident that it should have fairly wide uptake within the optical trapping community, at least those looking at more complex optical potentials. My one caveat is that as the idea is able to be applied to more general particle dynamics, I am unclear if there is an example of this algorithm being applied in an area separate to particle trapping, but I'm comfortable in stating that the claims main here look solid to me - faster, more flexible, more information. Certainly suitable for publication in Nature Comms.

I'm slightly embarrassed to say I have no real changes to suggest. One small one would be to note the region of interest used to be able to sample 4500 frames per second. I tried on my basler camera and couldn't quite get to that rate, so a comment on useful/minimum sampling frequencies would be good.

The only real concern I had was on the speckle field trapping where the 2D field is pushing the particle up towards the top of the sample - does it touch the top, and if so, how is this accounted for? I'd also imagine the particle, as it samples, the speckle field must move up and down, potentially moving out of focus - how is this accounted for?

Reviewer #3 (Remarks to the Author):

This manuscript proposed the use of a sequence of combinations of instantaneous position and velocity of a Brownian particle to retrieve the force field acting on the particle. Using the concept, the

authors also introduces a new algorithm for microscopic Force Reconstruction via Maximum-likelihood-estimator Analysis (FORMA) as a demonstration for force fields are calculated. These examples show FORMA is not only more accurate but also significantly more efficient (faster) than conventional means in determining the force fields in 1D and 2D problem. Combining instantaneous position and velocity pairs, FORMA has also the capability to mapping with the non-conservative forces – forces that cannot be determined by the particle positions along. These example show also the ability of identifying both the stable and unstable equilibrium points in multi-well potentials in 1D, and multiple stability points established by interference of coherent light sources in the form of speckles.

FORMA's advantage, as stated by the authors, arises from 1) simultaneous use of position and velocity, and 2) the efficiency of the well-developed linear regression algorithm. As far as I can tell, this is the first time anyone has proposed such concept, and at the same time, demonstrated in detail how such algorithm is used to determine a number of interesting force fields by a Brownian particles.

To take advantage of the efficiency of linear regression, the method would require sufficient spatial resolution of the imaging resolution of position tracking so the force field is linear in each region of interest. Presumably, for most force fields of interest to colloidal science and engineering, the spatial resolution is good enough. Nevertheless, it would be helpful, if the authors could add a succinct discussion in the manuscript.

In Figure 2, the (c & f) appears identical. The authors should double check if wrong figures were used here. Also, the mention that the computer (MacBook Air) was used to do the simulation does not seem relevant.

In Figure 3, the insert graphs in g, labeling with LG0 and LG2, seem identical. The authors should double check if wrong figures were used there.

Overall, this is a very interesting piece of research that teach how a combination of simultaneous positions and velocity of a Brownian particle can be used to map the force field, both conservative and non-conservative, single well and multiple wells in 1D and 2 D, vortex fields in 2D. It is easy to see how this concept and algorithm can be extended to applications in 3D. The illustration and the provided numerical algorithm will have a high impact to the practitioners in a broad community ranging from physics, biology and nano-photonics. I recommend it for publication in Nature Communications with minor revision.

Reviewer #1

Optical trapping remains an important field with continued success in a variety of areas in mesoscopic physics. One of the key requirements of the area is a determination of forces which is associated with a deeper understanding of the Brownian dynamics and optical fields interacting with the trapped object.

This manuscript develops a new algorithm for microscopic Force Reconstruction using a Maximum-likelihood-estimator (MLE) Analysis (FORMA) to retrieve the force field acting on a Brownian particle from the analysis of its displacements. This approach of FORMA allows the authors to make precise simultaneous estimations of both the conservative and non-conservative components of the force field. Importantly, even for the highest number of samples, the algorithm run time is in the order of a few ms on a laptop making this approach very attractive. The performance of FORMA is benchmarked with other established methods typically used in the calibration of optical tweezers the potential, PSD, and ACF analyses and studies performed with light fields with spin and orbital angular momentum and for particles in speckle for particles $\sim 1\mu\text{m}$ in diameter.

I would remark that non-conservative fields have come to the fore recently due to investigations in both standard traps and also for particles trapped in – for example – optical vortices where scattering of light can transfer orbital angular momentum.

Overall, this is an interesting, topical paper that is likely to find interest in the trapping community and potentially beyond. In parts it is well written and clear. Some questions for the authors

We thank this Reviewer for recognizing the interest and timeliness of our manuscript. We address his/her questions in detail below.

1) FORMA relies on the fact that in the proximity of an equilibrium position the force field can be approximated by a linear form leading to the use of a linear MLE. How accurate would this be if we move away dramatically from harmonic/Gaussian forms of traps or even have highly aberrated/distorted traps – e.g. a trap inside a cell? When does the assumptions made here breakdown?

We thank the Reviewer for their comment as it helps clarify a very important aspect of FORMA.

The fact that a generic force field can be approximated with a linear form in the proximity of an equilibrium position is quite general. This is a consequence of the fact that any function, no matter how complex, can always be approximated as a finite sum of higher-order terms. In particular,

any force field $\mathbf{F}(\mathbf{r}) = [F_x(x, y), F_y(x, y)]$ can be represented as a Taylor series expansion around a position $\mathbf{r}_0 = [x_0, y_0]$ as

$$\begin{bmatrix} F_x(x, y) \\ F_y(x, y) \end{bmatrix} = \underbrace{\begin{bmatrix} F_x(x_0, y_0) \\ F_y(x_0, y_0) \end{bmatrix}}_{\mathbf{F}_0} + \underbrace{\begin{bmatrix} \frac{\partial F_x}{\partial x}(x, y) & \frac{\partial F_x}{\partial y}(x, y) \\ \frac{\partial F_y}{\partial x}(x, y) & \frac{\partial F_y}{\partial y}(x, y) \end{bmatrix}}_{\mathbf{J}_0} \begin{bmatrix} x - x_0 \\ y - y_0 \end{bmatrix} + o(r), \quad (1)$$

where $o(r)$ contains the higher-order terms, $r = \sqrt{(x - x_0)^2 + (y - y_0)^2}$ and at an equilibrium point $\mathbf{F}_0 = \mathbf{0}$. We remark that this is true for any force field and, therefore, also for trapping potentials that are generated by aberrated or distorted beams.

There are two cases when the linear approximation breaks down:

1. *The linear approximation is strictly valid only around the equilibrium point.* This means that the force field seen by a particle around an equilibrium point is harmonic only if the particle does not move too much away from the equilibrium point. This depends on the intensity of the Brownian (thermal) noise and the depth of the potential. We remark that, even for the case of an optical trap generated by a non-aberrated Gaussian beam, the optical trap is harmonic only up to about half wavelength away from the trap center for particles significantly smaller than the light wavelength (corresponding to a few hundred nanometers), or up to about the particle diameter for particles whose size is comparable to the light wavelength; however, the trap is very well described by a harmonic potential because the particles only move a few tens to hundreds of nanometers away from the trap center for typical trap strengths. Therefore, already in a standard optical tweezers the harmonic form will depend mainly on the trap power: when the power is high enough we can always assume that the spatial region the particle explores corresponds to the one of a harmonic well – this is the most common case in practical situations; however, when the power of the laser beam is low, the non-linear behavior in the trap may become evident as a consequence of aberrations or simply of the Gaussian profile of the laser beam. In the manuscript, we already show that FORMA is indeed not restricted to harmonic potentials as it can be fairly easily adapted to correctly estimate far more complex situations, such as the double well potential in Fig. 4 and the speckle pattern in Fig. 5. The speckle pattern, being the result of the interference of many waves with random phases, is a particular interesting case of an extreme deviation from a harmonic potential. In many of these situations relevant in optics, the analysis can be performed by linearizing the force fields around a narrower spatial domain (or region of interest) where the linear approximation holds, and it can still be enough to properly reconstruct the overall force field as a consequence.

2. *The leading term in the Taylor expansion should be the linear term, i.e.,*

we must have $\mathbf{J}_0 \neq \mathbf{0}$. As long as $\mathbf{J}_0 \neq \mathbf{0}$, any force field near an equilibrium point can be approximated to first order by a linear form. This is not the case only in very special cases, typically where the optical potential is engineered not to have a harmonic component for some specific purposes. In this case, the linear form of the force field is insufficient, and one would need to adapt FORMA by truncating the approximation to a higher order and using a higher-order MLE estimator.

We now clarify these points in the text of the revised manuscript (pages 10-11):

“Finally, we applied it to more general force-field landscapes, including situations where the shape of the potential is not harmonic and the forces are too shallow to achieve long-term trapping: we used FORMA to identify the equilibrium points; to classify them as stable, unstable and saddle points; and to characterise their local force fields.”

and

“For force fields that cannot be approximated by a linear form around an equilibrium position because $\mathbf{J}_0 = \mathbf{0}$, FORMA needs to be extended using a higher-order MLE estimator.”

2) Can FORMA truly extract 3D information of a single particle or even be used for multiple trapped particles at different x,y,z positions? The authors seem to fleetingly state: FORMA can be straightforwardly extended also to measure 3D force fields.”

This is insufficient and needs data to support the 3D nature of the approach and some detailed discussion. It should be included to aid fair comparison to the other methods used in optical tweezers. At present this appears a shortcoming of the approach. See next point too which shows a method for 3D force field determination.

The equations for FORMA in 3D can be straightforwardly derived following the same steps as for FORMA in 2D. Specifically, the MLE of a particle in 3D near an equilibrium position can be explicitly written as

$$\mathbf{J}_0^* = \begin{bmatrix} J_{0,11}^* & J_{0,12}^* & J_{0,13}^* \\ J_{0,21}^* & J_{0,22}^* & J_{0,23}^* \\ J_{0,31}^* & J_{0,32}^* & J_{0,33}^* \end{bmatrix} = (\tilde{\mathbf{r}}^T \tilde{\mathbf{r}})^{-1} \tilde{\mathbf{r}}^T \mathbf{f}, \quad (2)$$

where, assuming to have N measurements,

$$\mathbf{f} = \begin{bmatrix} \gamma \frac{\Delta x_1}{\Delta t_1} & \gamma \frac{\Delta y_1}{\Delta t_1} & \gamma \frac{\Delta z_1}{\Delta t_1} \\ \dots & \dots & \dots \\ \gamma \frac{\Delta x_n}{\Delta t_n} & \gamma \frac{\Delta y_n}{\Delta t_n} & \gamma \frac{\Delta z_n}{\Delta t_n} \\ \dots & \dots & \dots \\ \gamma \frac{\Delta x_N}{\Delta t_N} & \gamma \frac{\Delta y_N}{\Delta t_N} & \gamma \frac{\Delta z_N}{\Delta t_N} \end{bmatrix} \quad (3)$$

and

$$\tilde{\mathbf{r}} = \begin{bmatrix} x_1 & y_1 & z_1 & 1 \\ \dots & \dots & \dots & \dots \\ x_n & y_n & z_n & 1 \\ \dots & \dots & \dots & \dots \\ x_N & y_N & z_N & 1 \end{bmatrix}. \quad (4)$$

The estimated particle diffusivity along each axis can be calculated from the residual error of the MLE

$$\begin{aligned} D_x^* &= \frac{1}{N} \sum_{n=1}^N \frac{\Delta t_n}{2\gamma^2} (f_{x,n} - J_{0,11}^* x_n - J_{0,12}^* y_n - J_{0,13}^* z_n)^2, \\ D_y^* &= \frac{1}{N} \sum_{n=1}^N \frac{\Delta t_n}{2\gamma^2} (f_{y,n} - J_{0,21}^* x_n - J_{0,22}^* y_n - J_{0,23}^* z_n)^2, \\ D_z^* &= \frac{1}{N} \sum_{n=1}^N \frac{\Delta t_n}{2\gamma^2} (f_{z,n} - J_{0,31}^* x_n - J_{0,32}^* y_n - J_{0,33}^* z_n)^2. \end{aligned} \quad (5)$$

To check that these formulas work, we have simulated the motion of an optically trapped particle in 3D with both conservative and non-conservative components along all axes. Specifically, we provide the results of the analysis with FORMA of a 3D Brownian simulation, assuming a harmonic trap with different k 's and Ω 's along the three coordinate axes. We used a spherical particle with radius $R = 0.5 \mu\text{m}$ in an aqueous medium of viscosity $\eta = 0.0011 \text{ Pa s}$ and $D = 0.392 \mu\text{m}^2\text{s}^{-1}$ at a sampling frequency $f_s = 4504.5 \text{ s}$, and we used 24 windows of $2 \cdot 10^5$ samples for the averaging. These results show very good agreement between the k 's and Ω 's imposed in the simulations and reconstructed by FORMA in 3D, as shown by the table below corresponding to Supplementary Table 1 in the revised manuscript.

coord.	k (pN μm^{-1})	k^* (pN μm^{-1})	Ω (s^{-1})	Ω^* (s^{-1})	D ($\mu\text{m}^2 \text{s}^{-1}$)	D^* ($\mu\text{m}^2 \text{s}^{-1}$)
x	0.6	0.60 ± 0.02	2	2.27 ± 1.15	0.392	0.387 ± 0.001
y	0.5	0.49 ± 0.02	1.5	1.57 ± 0.81	0.392	0.387 ± 0.002
z	0.2	0.20 ± 0.01	3	2.86 ± 0.78	0.392	0.389 ± 0.001

Supplementary Table 1: Estimated parameters in a non-conservative 3D trap using FORMA. Results of the analysis with FORMA of a 3D Brownian simulation, assuming a harmonic trap with different k 's and Ω 's along the three coordinate axes. We simulated a spherical particle with radius $R = 0.5 \mu\text{m}$ in an aqueous medium of viscosity $\eta = 0.0011 \text{ Pa s}$ at a sampling frequency $f_s = 4504.5 \text{ s}$, and we used 24 windows of $2 \cdot 10^5$ samples for the averaging.

In the revised version of the manuscript we have added the explicit equations of FORMA in 3D (see section ‘‘FORMA in 3D’’ in the Methods of

the revised manuscript) and additional simulations to demonstrate that FORMA works also in 3D (see Supplementary Table 1).

3) The following paper shows relevant work and is not cited nor discussed in context. The paper extracts local forces from trajectories of an optically trapped particle, and reveals the three-dimensional force field experienced by a Rayleigh particle with 10 nm spatial resolution and femtonewton precision in force.

Direct Measurement of the Nonconservative Force Field Generated by Optical Tweezers Pinyu Wu, Rongxin Huang, Christian Tischer, Alexandr Jonas, and Ernst-Ludwig Florin *Phys. Rev. Lett.* 103, 108101 (2009).

We thank the Reviewer for drawing our attention to this paper, which is certainly important in the context of this manuscript. The method presented in the suggested paper is based on a 0th-order approximation of the local force, so it assumes that in a small volume the force is constant. This is quite a strong assumption and requires the volumes to be very small to ensure that the force is sufficiently constant within them. This also means that a lot of data points are needed in order to sample the forces with sufficient statistics. Obtaining so many data points can be very challenging or impossible in several situations, namely: (1) in the proximity of unstable equilibria, where the probe particles tend not to linger for too long; (2) for complex potential landscapes (such as those produced by speckle fields), where the phase space to explore is too large; and (3) in slowly time-varying systems, where the time for acquiring data is limited. All these cases can be dealt with using FORMA thanks to the fact that it is based on a 1st-order approximation of the force field, and therefore requires less data and can use larger acquisition areas.

We now cite and discuss this paper in the revised manuscript (page 4):

“Ref. [13] has proposed a direct force measurement method based on taking the average of f_n around each position of the force field; this approach requires many more samples than FORMA.”

[13] Wu, P., Huang, R., Tischer, C., Jonas, A. & Florin, E.-L. Direct measurement of the nonconservative force field generated by optical tweezers. *Phys. Rev. Lett.* 103, 108101 (2009).

5) There are several English grammar that might be clarified and errors in references “the precise angular-momentum-transfer mechanisms can be very complex when the beam is focalized.” – should be “...when the beam is tightly focused” e.g. ref 13, 19 seem incomplete;

Figure 3: if the y-axis is angular velocity should it not be rad/s and not just per second?

We thank the Reviewer for pointing out these spelling mistakes. We have

now corrected the manuscript. We have also thoroughly revised the whole manuscript by performing a spell-check. Also, we now speak of angular frequency instead of angular velocity.

6) is there any drawback to using non-spherical particles or Rayleigh particles with FORMA? All data presently shown is for larger spherical objects.

We thank the Reviewer for this observation. FORMA will work as explained in the manuscript for spherical particles of any size, including Rayleigh particles. For non-spherical particles, FORMA needs to be extended in order to account both for the translational and rotational degrees of freedom of the particles so that one would need to have a version of FORMA in 6 dimensions in the most general case; in such cases, the main challenges would be to track such degrees of freedom with sufficient accuracy and statistics.

We now remark these points in the discussion of the revised manuscript (page 10):

“Even though we have presented results only for particles of a single size, FORMA can be used as described in this work for spherical particles of any size, from Rayleigh particles much smaller than the optical wavelength to large Mie particles. [...] FORMA can also be extended to deal with non-translational degrees of freedom, which might be important when dealing with non-spherical particles.”

7) The authors state: “FORMA can be applied in situations that require a fast response such as in real-time applications and in the presence of time-varying conditions.” Is there any such specific situation they can show where FORMA works but the other methods do not?

We thank the Reviewer for this question that permits us to dig deeper into this important advantage of FORMA. With this statement we want to highlight how the fact that FORMA is faster and requires less data will potentially open opportunities for new kinds of experiments and setups. Here we provide some examples. First, measurements done within biological systems (e.g., measurements performed within the cytoplasm of cells, measurements performed in bacterial baths) often suffer from drifts due to changes of the physical and chemical conditions of the environment. Second, nanothermodynamic measurements probing the properties of systems out of equilibrium often need to take into account changes of the environment over time due to, e.g., changes of temperature. Third, haptic systems that rely on the continuous interaction with an external user often have strict constraints on the timescales during which calibration should be performed in order not to expose the user to delays. In all these cases, being able to calibrate the optical tweezers on a timescale one or two order of magnitude faster than traditional methods can be critical for the feasibility of experiments.

We now highlight this in the revised manuscript (page 10):

“[...] requiring less data and having a faster run time, FORMA can be applied to situations that require a fast response such as in real-time applications (e.g., in haptic optical tweezers [31]) and in the presence of time-varying conditions (e.g., in the study of biological systems [32–34], active baths [35, 36] and of systems out of equilibrium [37–39]).”

[31] Pacoret, C. & Régnier, S. A review of haptic optical tweezers for an interactive microworld exploration. *Rev. Sci. Instrumen.* 84, 081301 (2013).

[32] Perkins, T. T. Optical traps for single molecule biophysics: A primer. *Laser Photon. Rev.* 3, 203–220 (2009).

[33] Ou-Yang, H. D. & Wei, M.-T. Complex fluids: Probing mechanical properties of biological systems with optical tweezers. *Annu. Rev. Phys. Chem.* 61, 421–440 (2010).

[34] Stevenson, D. J., Gunn-Moore, F. J. & Dholakia, K. Light forces the pace: optical manipulation for biophotonics. *J. Biomed. Opt.* 15, 041503 (2010).

[35] Maggi, C. et al. Generalized energy equipartition in harmonic oscillators driven by active baths. *Phys. Rev. Lett.* 113, 238303 (2014).

[36] Argun, A. et al. Non-Boltzmann stationary distributions and nonequilibrium relations in active baths. *Phys. Rev. E* 94, 062150 (2016).

[37] Seifert, U. Stochastic thermodynamics, fluctuation theorems and molecular machines. *Rep. Prog. Phys.* 75, 126001 (2012).

[38] Martínez, I. A., Rolán, É., Dinis, L. & Rica, R. A. Colloidal heat engines: A review. *Soft Matter* 13, 22–36 (2017).

[39] Pinçe, E. et al. Disorder-mediated crowd control in an active matter system. *Nat. Commun.* 7, 10907 (2016).

8) I don't get any sense of the limits of FORMA at present. For example the determination of stiffness using the power spectrum typically requires a harmonic potential, a sampling bandwidth well in excess of the trap frequency, a detailed knowledge of the drag coefficient, and some form of accurate position calibration. The authors have focused on speed in the paper rather than other aspects of FORMA. They should expand this aspect.

We thank the Reviewer for raising this point, which permits us to clarify the main advantages of FORMA beyond its speed as well as to discuss its requirements.

Beyond its speed, the main advantages of FORMA are:

1. it requires less data and therefore it converges faster and with smaller error bars for a given amount of experimental samples.
2. it has less stringent requirements on the input data, as it does not require a series of particle's positions sampled at regular time intervals or for a time long enough to reconstruct the equilibrium distribution.
3. it is simpler to execute and automatise because it does not have any analysis or fitting parameter to be chosen.
4. it probes simultaneously the conservative and non-conservative components of the force field.
5. since it does not need to use the trajectory of a particle held in a potential, it can identify and characterise both stable and unstable equilibrium points in extended force fields, and therefore it is compatible with a broader range of possible scenarios where a freely-diffusing particle is used as a tracer, e.g., in microscopy and rheology.
6. As we explain in more detail in the answer to question 1 by this Reviewer, FORMA can be used for general force fields with a shape not necessarily harmonic.

Thanks to all these characteristics, FORMA outperforms currently available optical tweezers calibration methods, and it greatly extends the situations that can be analyzed to, e.g., the characterization of unstable and metastable equilibrium points. This information is provided in the introduction of the manuscript (pages 2-3):

“This presents several advantages over the methods mentioned above. First, FORMA executes much faster because the algorithm is based on linear algebra for which highly optimised libraries are readily available. Second, it requires less data and therefore it converges faster and with smaller error bars. Third, it has less stringent requirements on the input data, as it does not require a series of particle's positions sampled at regular time intervals or for a time long enough to reconstruct the equilibrium distribution. Fourth, it is simpler to execute and automatise because it does not have any analysis parameter to be chosen. Fifth, it probes simultaneously the conservative and non-conservative components of the force field. Finally, since it does not need to use the trajectory of a particle held in a potential, it can identify and characterise both stable and unstable equilibrium points in extended force fields, and therefore it is compatible with a broader range of possible scenarios where a freely-diffusing particle is used as a tracer, e.g., in microscopy and rheology.”

The main requirement of FORMA is a bandwidth in excess (\sim one order of magnitude) of the characteristic times of the particle motion in the force

Supplementary Figure 2: Performance of FORMA as a function of sampling time. Values obtained by FORMA from simulated timeseries of an optically trapped particle for (a) the trap stiffness k and (b) the diffusion coefficient D as function of the sampling frequency f . The vertical blue and red lines respectively indicate the characteristic frequency in the trap ($f_c = k/\gamma = 60.3 \text{ s}^{-1}$) and the sampling frequency ($f_s = 4504.5 \text{ s}^{-1}$) of the results reported in Figure 2. The shaded areas represent the regions where the error in the determinations is greater than 10%. In all cases, we have simulated 24 trajectories of the motion of a spherical microparticle with radius $R = 0.5 \mu\text{m}$ in an aqueous medium of viscosity $\eta = 0.0011 \text{ Pa s}$. The time for each simulations is fixed to 22 s.

field to be probed, which is analogous to what happens with other methods such as ACF and PSD. To address this point more quantitatively, we have performed new simulations to test the accuracy of the reconstruction algorithm as a function of the sampling rate. The corresponding data are shown in Supplementary Figure 2 in the revised version of the manuscript. This figure shows the plots of the stiffness k and diffusion D obtained with FORMA as a function of the data sampling rate using the same physical parameters in the Brownian simulations as in the experimental data shown in Fig. 1 of the manuscript. We can see that the method provides accurate results, with an error less than 10%, when the sampling frequency is ten-fold larger than the characteristic frequency of the trap (dashed blue line). The dashed red line points out the sampling rate used in the experiments ($f_s = 4504.5 \text{ s}^{-1}$). We now discuss this in the revised manuscript (page 3):

“Note that, as in the case of the PSD and ACF analyses, the sampling frequency needs to be at least about one order of magnitude greater than the characteristic trap frequency in order to obtain accurate results (see also Supplementary Figure 2).”

Overall, I think there may be potential for this work for publication in Nature Communications but the authors first need to provide a detailed

response to the above issues.

We thank the Reviewer again for recognizing the potential of our work and we hope that we have addressed all of his/her concerns so that he/she will find our work suitable for publication in Nature Communications.

Reviewer #2

This is a nice piece of work and I'm confident that it should have fairly wide uptake within the optical trapping community, at least those looking at more complex optical potentials. My one caveat is that as the idea is able to be applied to more general particle dynamics, I am unclear if there is an example of this algorithm being applied in an area separate to particle trapping, but I'm comfortable in stating that the claims main here look solid to me - faster, more flexible, more information. Certainly suitable for publication in Nature Comms.

We thank the Reviewer for his/her positive view on our work and its suitability for Nature Communications.

I'm slightly embarrassed to say I have no real changes to suggest. One small one would be to note the region of interest used to be able to sample 4500 frames per second. I tried on my basler camera and couldn't quite get to that rate, so a comment on useful/minimum sampling frequencies would be good.

We thank the Reviewer to encourage us to clarify this important aspect of our method. In order to obtain accurate results, FORMA requires a sampling bandwidth in excess (\sim one order of magnitude) of the characteristic times of the particle motion in the force field to be probed. Importantly, this requirement is analogous to what happens with other methods such as ACF and PSD. To address this point more quantitatively, we have performed new simulations to test the accuracy of the reconstruction algorithm as a function of the sampling rate. The corresponding data are shown in Supplementary Figure 2 in the revised version of the manuscript (reported also in reply to question 8 of Reviewer 1). This figure shows the plots of the stiffness k and diffusion D obtained with FORMA as a function of the data sampling rate using the same physical parameters in the Brownian simulations as in the experimental data shown in Fig. 1 of the manuscript. We can see that the method provides accurate results, with an error less than 10%, when the sampling frequency is tenfold larger than the characteristic frequency of the trap (dashed blue line). The dashed red line points out the sampling rate used in the experiments ($f_s = 4504.5 \text{ s}^{-1}$). We now discuss this in the revised manuscript (page 3):

“Note that, as in the case of the PSD and ACF analyses, the sampling frequency needs to be at least about one order of magnitude greater than the characteristic trap frequency in order to obtain accurate results (see also Supplementary Figure 2).”

We would also like to remark that the sampling rate $f_s = 4504.5 \text{ s}^{-1}$ was achieved in the experiment by using a small region of interest (ROI,

corresponding to 256×40 pixels) in the CMOS camera and an exposure time of $100 \mu\text{s}$ in the acquisition of the videos. To make this point clearer in the manuscript, we added the information about the ROI and exposure time to the caption of Supplementary Fig. S1.

The only real concern I had was on the speckle field trapping where the 2D field is pushing the particle up towards the top of the sample - does it touch the top, and if so, how is this accounted for? I'd also imagine the particle, as it samples the speckle field must move up and down, potentially moving out of focus - how is this accounted for?

As the Reviewer points out, in our experimental configuration the probe bead is pushed towards the upper cover slip (without touching it because of electrostatic repulsion), where it remains confined in a quasi-2D configuration. The main effect of this proximity to the coverslip is that the value of the particle drag coefficient increases when compared to the case in bulk. More in detail, the particle position below the surface is determined by the equilibration of three forces:

1. *Buoyancy.* Since the particle is made of polystyrene, its density ($\rho_{\text{particle}} = 1050 \text{ kg m}^{-3}$) is only slightly higher than that of water ($\rho_{\text{water}} = 1000 \text{ kg m}^{-3}$) making its buoyancy almost neutral, providing a slight push downwards.
2. *Screened electrostatics.* This is due to the screened electrostatic repulsion between the particle and the coverslip. This force increases exponentially as the particle moves closer to the surface.
3. *Radiation pressure.* This force is due to the radiation pressure of the laser beam and pushes the particle up towards the coverslip. This force can depend on whether the particle is in a more or less illuminated area of the speckle. However, since the electrostatic repulsion increases exponentially as the particle approaches the surface, the equilibrium position reached by the particle is not significantly affected by this (specifically, thanks to this exponential dependence, increasing the optical force by a factor of 2 only changes the vertical position by a few nanometers). In fact, we verified from the videos that the vertical particle position does not change significantly during the experiments and that the particle never moves out of focus.

We now make these points clearer in the revised version of the manuscript (page 12):

“Because of the radiation pressure of the optical forces generated by the speckle light field, the probe bead is pushed towards the upper cover slip (without touching it because of screened electrostatic repulsion), where it remains confined in a quasi-2D configuration and diffuses exploring a wide area.”

Reviewer #3

This manuscript proposed the use of a sequence of combinations of instantaneous position and velocity of a Brownian particle to retrieve the force field acting on the particle. Using the concept, the authors also introduces a new algorithm for microscopic Force Reconstruction via Maximum-likelihood-estimator Analysis (FORMA) as a demonstration for force fields are calculated. These examples show FORMA is not only more accurate but also significantly more efficient (faster) than conventional means in determining the force fields in 1D and 2D problem. Combining instantaneous position and velocity pairs, FORMA has also the capability to mapping with the non-conservative forces – forces that cannot be determined by the particle positions alone. These example show also the ability of identifying both the stable and unstable equilibrium points in multi-well potentials in 1D, and multiple stability points established by interference of coherent light sources in the form of speckles.

FORMA's advantage, as stated by the authors, arises from 1) simultaneous use of position and velocity, and 2) the efficiency of the well-developed linear regression algorithm. As far as I can tell, this is the first time anyone has proposed such concept, and at the same time, demonstrated in detail how such algorithm is used to determine a number of interesting force fields by a Brownian particle.

To take advantage of the efficiency of linear regression, the method would require sufficient spatial resolution of the imaging resolution of position tracking so the force field is linear in each region of interest. Presumably, for most force fields of interest to colloidal science and engineering, the spatial resolution is good enough. Nevertheless, it would be helpful, if the authors could add a succinct discussion in the manuscript.

We thank the Reviewer for their comment as it helps clarify a very important aspect of our work. A spatial resolution significantly smaller than the characteristic length of the force field is necessary in order to obtain accurate results with all calibration methods, including FORMA. For colloidal experiments, such as those usually performed in optical trapping and optical manipulation, this translates to a requirement of a resolution of about 10 nm or better, which is nowadays routinely achieved by established digital video microscopy techniques.

We also remark that another important aspect is the time resolution, which enables us, in the case of FORMA, to fulfill the linear regression model shown in Eq. 1. We have discussed this in detail in the reply number 8 to Reviewer 1, and in the associated simulations shown in Supplementary Figure 2 in the revised version of the manuscript (reported also in reply to

question 8 of Reviewer 1).

We now remark these points in the revised manuscript (page 3):

“We have tracked the particle position using digital video microscopy [11] with a spatial resolution below 5 nm and with a frame frequency $f_s = 4504.5 \text{ s}^{-1}$, corresponding to a sampling timestep $\Delta t = 0.222 \text{ ms}$. Note that, as in the case of the PSD and ACF analyses, the sampling frequency needs to be at least about one order of magnitude greater than the characteristic trap frequency in order to obtain accurate results (see also Supplementary Figure 2).”

In Figure 2, the (c and f) appears identical. The authors should double check if wrong figures were used here. Also, the mention that the computer (MacBook Air) was used to do the simulation does not seem relevant.

We have double-checked these figures and they are correct. Figs. 2c and 2f do look indeed quite similar, but in fact they are different (compare for example their initial values). The similarity is because the computing time depends mainly on the number of data points and not on the physical values of the computation.

Following the Reviewer’s suggestion, we have removed the reference to the brand of the laptop but we kept the main specifications in terms of CPU and RAM.

In Figure 3, the insert graphs in g, labeling with LG0 and LG2, seem identical. The authors should double check if wrong figures were used there.

We have double-checked the insets in Fig. 3g and they are correct. The difference is minimal and the torque only appears as a mild bending of the force vectors around the equilibrium point that can be very difficult to distinguish by eye. We have re-drawn these insets to make the differences more clear and we now note in the figure caption that “The insets in (g) show the force fields for the case of a Gaussian beam (LG₀), which is purely conservative ($l = 0$), and of a beam with a charge $l = 2$ of orbital angular momentum (LG₂), which features a non-conservative component that only induces a mild bending of the arrows.”

Overall, this is a very interesting piece of research that teach how a combination of simultaneous positions and velocity of a Brownian particle can be used to map the force field, both conservative and non-conservative, single well and multiple wells in 1D and 2 D, vortex fields in 2D. It is easy to see how this concept and algorithm can be extended to applications in 3D. The illustration and the provided numerical algorithm will have a high impact to the practitioners in a broad community ranging from physics, biology and

nano-photonics. I recommend it for publication in Nature Communications with minor revision.

We thank the Reviewer for acknowledging the interest and potential impact of this work, and for recommending publication in Nature Communications.

REVIEWERS' COMMENTS:

Reviewer #1 (Remarks to the Author):

The revision of the paper is thorough and comprehensive. It clearly answers my points. I am pleased to recommend publication in Nature Communications

Reviewer #2 (Remarks to the Author):

The authors have done a very thorough job in addressing the comments made by myself and the author referees. It looks to be a tighter and clearer manuscript now and I continue with my recommendation for publication in Nature Communications.

Reviewer #3 (Remarks to the Author):

I have read the rebuttal by the authors to the reviewers comments. I am satisfied with the answers and changes made by the authors. I have no more comments except to recommend the manuscript be published.

Manuscript NCOMMS-18-26700A

Authors' response to Reviewers' comments

Reviewer #1

The revision of the paper is thorough and comprehensive. It clearly answers my points. I am pleased to recommend publication in Nature Communications.

We thank the Reviewer and we are happy to have addressed satisfactorily all points.

Reviewer #2

The authors have done a very thorough job in addressing the comments made by myself and the other referees. It looks to be a tighter and clearer manuscript now and I continue with my recommendation for publication in Nature Communications.

We thank the Reviewer and we are happy to have addressed satisfactorily all points.

Reviewer #3

I have read the rebuttal by the authors to the reviewers comments. I am satisfied with the answers and changes made by the authors. I have no more comments except to recommend the manuscript be published.

We thank the Reviewer and we are happy to have addressed satisfactorily all points.